# Cooperative Meta-Learning with Gradient Augmentation

**Jongyun Shin**[1]        **Seungjin Han**[1]        **Jangho Kim**[*1]

[1]Computer Science Department, Kookmin University, Seoul, Korea
`{whddbs519,gkstmdwls99,jangho.kim}@kookmin.ac.kr`

## Abstract

Model agnostic meta-learning (MAML) is one of the most widely used gradient-based meta-learning, consisting of two optimization loops: an inner loop and outer loop. MAML learns the new task from meta-initialization parameters with an inner update and finds the meta-initialization parameters in the outer loop. In general, the injection of noise into the gradient of the model for augmenting the gradient is one of the widely used regularization methods. In this work, we propose a novel cooperative meta-learning framework dubbed CML which leverages gradient-level regularization with gradient augmentation. We inject learnable noise into the gradient of the model for the model generalization. The key idea of CML is introducing the co-learner which has no inner update but the outer loop update to augment gradients for finding better meta-initialization parameters. Since the co-learner does not update in the inner loop, it can be easily deleted after meta-training. Therefore, CML infers with only meta-learner without additional cost and performance degradation. We demonstrate that CML is easily applicable to gradient-based meta-learning methods and CML leads to increased performance in few-shot regression, few-shot image classification and few-shot node classification tasks. Our codes are available at `https://github.com/JJongyn/CML`.

## 1 INTRODUCTION

Meta-learning, also known as "learning to learn", is a methodology to learn a new task by utilizing previous

---

\* Corresponding Author

knowledge and experience [Vilalta and Drissi, 2002]. Model-agnostic meta-learning (MAML) [Finn et al., 2017] is one of the dominant gradient-based meta-learning methods [Rajeswaran et al., 2019, Rusu et al., 2019, Gupta et al., 2020]. MAML consists of two optimization loops including an inner loop and an outer loop. The inner loop adapts the model with task-specific knowledge and the outer loop finds the meta-initialization parameters which can quickly adapt the new task knowledge in the inner loop, called task-adaptation. Generally, meta-learning with a few-shot setting involves both meta-training and meta-testing. In meta-training, a variety of few-shot learning tasks are provided for a meta-learner and the meta-learner should solve an unseen task with few-shot samples in meta-testing. In the process, meta-learner learns the ability to adapt to various tasks, but they are challenged to form meta-initialization parameters with well-generalized knowledge.

Traditionally, noise injection to the model is widely used for improving the generalization performance of the model. Neelakantan et al. [2015] finds that adding noise to a network's gradient improves the network's generalization performance. Similarly, Yang et al. [2020] performs gradient augmentation by pruning the model to create multiple sub-networks and using different data augmentations for each input in the sub-networks for inducing the diversity into the gradient, but this requires multiple inferences. They show that injecting noise into gradients plays an important role in improving generalization performance.

Motivated by the regularization effect of noise in gradients and the diverse gradient augmentation for the model generalization, we propose a novel cooperative meta-learning (CML) framework. It can be applied with gradient-based meta-learning to find better meta-initialization parameters through a regularization effect but has no additional cost at test time. CML has three parts which are the feature extractor, meta-learner and co-learner. The feature extractor and meta-learner parameters already exist in the original MAML and co-learner is newly introduced in this work for generating the new gradient. The co-learner is a plug-and-

*Accepted for the 40th Conference on Uncertainty in Artificial Intelligence* (UAI 2024).

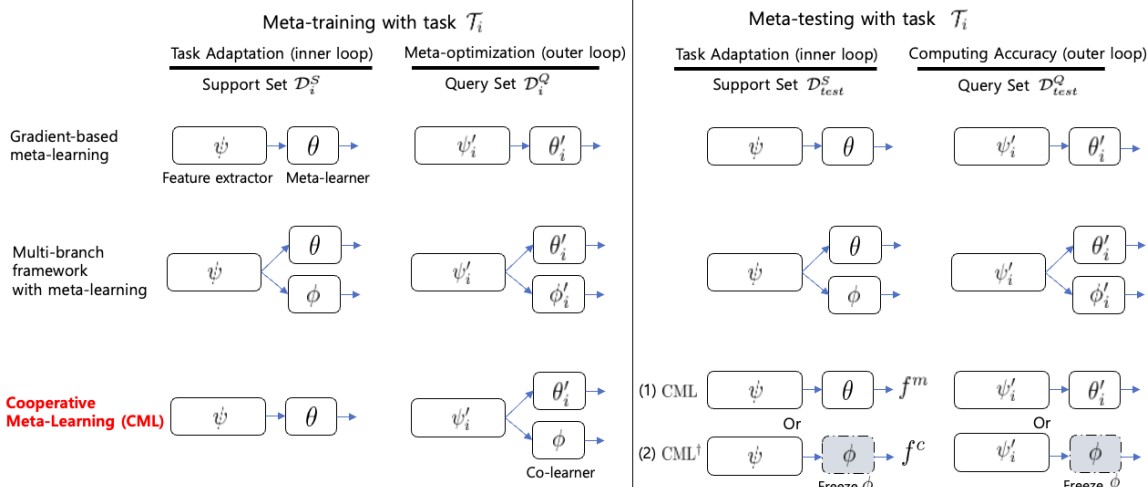

Figure 1: Overall process of CML and comparisons with other methods with a given task ($\mathcal{T}_i$). $\psi$, $\theta$ and $\phi$ denote meta-initialization parameters of the feature extractor, meta-learner and co-learner. The feature extractor $\psi$ extracts the features, i.e., body layers of DNN. The meta-learner $\theta$ and co-learner $\phi$ predict outputs based on the features, i.e., classifier. $\psi_i'$, $\theta_i'$, and $\phi_i'$ means adapted parameters with $i$-task during an inner loop. Since CML does not adapt the co-learner to the task for generalization from gradient augmentation, after meta-training, CML can infer without additional costs. In meta-testing, CML evaluates performance after performing a task-adaptation, like standard MAML having $\psi$ and $\theta$. On the other hand, CML$^\dagger$ has parameters $\psi$ and $\phi$, where only $\psi$ performs the task-adaptation and then evaluates the performance.

play module that takes the features of the feature extractor as input and generates their gradients by backpropagation. Thus, its goal is to provide a gradient for augmentation from a different perspective than the meta-learner, creating an augmented meta-gradient. We think that this is effective as a learned meaningful noise generated by the training of the co-learner rather than simply adding random noise. To achieve our goal, we design the CML with two purposes: Firstly, the co-learner arouses a different point of view from the naive meta-learner for generalization ability and diversity of meta-gradient. Secondly, the co-learner can be easily deleted at test time without any accuracy drop, which means the co-learner affects only finding meta-initialization parameters not learning a new task.

Figure 1 shows the overall process of CML and comparisons with other methods such as a naive gradient-based meta-learning and a multi-branch framework with meta-learning. In meta-training, our newly introduced co-learner is only updated in the outer loop, which means the meta-learner solely adapts the new task in the inner loop. Since the co-learner is updated at the previous outer loop, not the current inner loop, the co-learner cooperatively finds the meta-initialization parameters of the shared feature extractor by gradient augmentation in the meta-gradient with a different perspective than the meta-learner. Hence, CML does not need to make a sub-network such as pruning and use a different data augmentation with multiple inferences for the diversity. Also, after meta-training, CML can easily delete the co-learner because the co-learner does not change the

meta-initialization parameters in the inner loop. Therefore, CML can only infer the feature extractor and meta-learner in meta-testing. Another variation of CML, the co-learner without task-adaptation can be used with the feature extractor which is represented as CML$^\dagger$ in Figure 1. Our main contributions are summarized as follows:

- We propose the cooperative meta-learning (CML) framework which finds the better meta-initialization parameters without additional cost at test time. Unlike previous regularization methods, our proposed co-learner generates diverse meta-gradient without multiple data augmentation, inference and pruning.

- We verify the effectiveness of CML and its applicability, where CML is applied with gradient-based meta-learning methods on various tasks such as few-shot regression, few-shot image classification and few-shot node classification tasks.

- We show that CML's gradient augmentation induces gradient diversity and conduct an analysis of the gradient of the co-learner and meta-learner during meta-optimization.

- We demonstrate that the performance improvement is not solely attributed to the additional parameters of the co-learner during meta-training, but rather to the framework of CML with meta-training.

## 2 RELATED WORK

### 2.1 GRADIENT-BASED META-LEARNING

In recent, meta-learning successfully covers a diverse application [Hospedales et al., 2021]. Gradient-based meta-learning optimizes a bilevel optimization problem [Colson et al., 2007] where it has a task-adaptation (inner loop) learning a new task with a few shot samples from meta-initialization parameters and meta-optimization (outer loop) finding proper meta-initialization parameters from an inner loop update. Many variants of MAML have been studied in various domains [Yin et al., 2020, Obamuyide and Vlachos, 2019, Collins et al., 2022, Lee et al., 2021]. BOIL [Oh et al., 2020] tackles the feature reuse problem in meta-optimization and freezes the classifier in task-adaptation. Sharp-MAML [Abbas et al., 2022] leverages sharpness-aware minimization to solve a bilevel optimization problem. In this work, we propose a new meta-learning framework that can be applied to any gradient-based meta-learning.

### 2.2 MULTI-BRANCH FRAMEWORK

While maintaining the exact computational graph for inference, many works to boost the performance of the model have been studied. Auxiliary training adds auxiliary classifiers connected in intermediate layers [Szegedy et al., 2015, Zhang et al., 2020] and multi-task learning simultaneously learns multiple related tasks and the knowledge from multi-task can be reused by the others [Yang and Hospedales, 2017]. Unlike previous methods, multi-branch frameworks [Kim et al., 2021, Xie and Du, 2022, Liang et al., 2022] shared intermediate layers and split multi-branch under the same task which utilizes knowledge distillation [Hinton et al., 2015] transferring the knowledge to enhance independent branches. Zhu et al. [2018] split the model into several sub-networks and made an ensemble logit to teach individual sub-networks. Song and Chai [2018] introduces multiple heads from the same network to improve the generalization of the model.

### 2.3 REGULARIZATION BY NOISE

To improve the generalization performance, various ways to impose constraints on model structure and gradients by noise have been studied. Hinton and Roweis [2002] uses gaussian gradient noise schedule to train the embedding model. Dropout [Srivastava et al., 2014] randomly drops the connections during training which introduces the random noise into forward propagation. Similarly, Huang et al. [2016] randomly disconnects the layers during training. Neelakantan et al. [2015] shows injecting noise to gradient works very deep architecture. GradAug [Yang et al., 2020] generates meaningful noise in gradients rather than

random noises by multiple data augmentation and pruning of the model.

Cooperative meta-learning leverages the advantages of both multi-branch framework and regularization by noise in the gradient-based meta-learning domain. CML only introduces a co-learner and trains it in meta-optimization to augment a meta-gradient by sharing the feature extractor. It induces a regularization effect by injecting noise into the meta-gradient without multiple forwarding or making a sub-network such as pruning.

## 3 METHODOLOGY

In this section, we give a brief explanation of the Model-Agnostic Meta-Learning (MAML) and then, we explain a proposed cooperative meta-learning (CML) framework which is an extension of MAML that uses cooperative learning with gradient augmentation to learn meta-initialization parameters of the DNN. In meta-learning, the ability to generalize to a new task is a challenging problem. To solve this problem, we introduce a co-learner that drives the augmentation at the gradient-level regularization.

### 3.1 MODEL-AGNOSTIC META-LEARNING (MAML)

In this work, we divide the DNN model used for meta-learning into two groups: the feature extractor $\psi$ which extracts the features, i.e., body layers of DNN and the meta-learner $\theta$ predicting outputs based on the features, i.e., classifier. We sample a set of tasks $\{\mathcal{T}\}_i^N$ containing N tasks from the task distribution $p(\mathcal{T})$. DNN model represented by a $f_{(\psi,\theta)}$ is trained using samples from each task $\mathcal{T}_i$ under the two optimization loops. These samples $\mathcal{D}_i$ are divided into support set $\mathcal{D}_i^S$ and query set $\mathcal{D}_i^Q$ which are used in the inner loop and outer loop, respectively. MAML consisting of two optimization loops which are the inner loop and outer loop tries to find well-generalized meta-initialization parameters during meta-training. In the inner loop, we update task-specific parameters from meta-initialization parameters $(\psi, \theta)$ using the support set with an outer step size of $\alpha$.

$$(\psi_i', \theta_i') = (\psi, \theta) - \alpha \nabla_{(\psi,\theta)} \mathcal{L}(f_{(\psi,\theta)}; \mathcal{D}_i^S) \quad (1)$$

and takes totally $M$-updates for task-specific parameters.

$$(\psi_i', \theta_i') \leftarrow (\psi_i', \theta_i') - \alpha \nabla_{(\psi_i',\theta_i')} \mathcal{L}(f_{(\psi_i',\theta_i')}; \mathcal{D}_i^S) \quad (2)$$

We will consider one gradient step for the rest for simplification. After task-adaptation in the inner loop, we compute each task loss for the query set with task-specific parameters $(\psi_i', \theta_i')$. By summing all task losses, meta-optimization

optimizes the following objectiveness:

$$\min_{\psi,\theta} \sum_i^N \mathcal{L}(f_{(\psi_i',\theta_i')}; \mathcal{D}_i^Q) =$$
$$\sum_i^N \mathcal{L}(f_{(\psi,\theta)-\alpha\nabla_{(\psi,\theta)}\mathcal{L}(f_{(\psi,\theta)};\mathcal{D}_i^S)}; \mathcal{D}_i^Q) \quad (3)$$

In the outer loop, we update meta-initialization parameters with $N$ task losses using meta-gradient by meta-optimization with an outer step size of $\beta$.

$$(\psi,\theta) \leftarrow (\psi,\theta) - \beta\nabla_{(\psi,\theta)} \sum_i^N \mathcal{L}(f_{(\psi_i',\theta_i')}; \mathcal{D}_i^Q) \quad (4)$$

In meta-testing, we verify the trained meta-initialization parameters. The inner loop adapts to the new task with a support set that remains the same as in meta-training. However, the outer loop only computes the accuracy using a query set for each task. There is no meta-optimization process in the outer loop of meta-testing.

## 3.2 COOPERATIVE META-LEARNING (CML)

Our proposed framework includes an additional module called co-learner $\phi$ inducing gradient augmentation in the meta-optimization. The co-learner can consist of a convolution layer or a fully connected layer, depending on the task. In meta-training, CML performs task-adaptation with the feature extractor and meta-learner in the inner loop same as the original gradient-based meta-learning such as MAML. The co-learner is added to the feature extractor during meta-optimization in the outer loop. Note that the co-learner only intervenes in the outer loop to perform meta-optimization with the feature extractor and meta-learner. In other words, the co-learner does not perform task-adaptation for the current task in the inner loop, therefore, it has implicit knowledge of tasks in the previous sampled batch. As a result, it has a different representation from the naive meta-learner. In this framework, the meta-learner and co-learner always share a representation of the feature extractor. Hence, their gradients are aggregated in the feature extractor, resulting in gradient augmentation.

Formally, we sample a support set $\mathcal{D}_i^S$ and query set $\mathcal{D}_i^Q$ from a new task $\mathcal{T}_i$. Also, we denote the initial parameters for the feature extractor, meta-learner and co-learner as $\psi, \theta$ and $\phi$ and two models in our framework: model $f^m$, which composes of a shared feature extractor and a meta-learner, and $f^c$, which composes of a shared feature extractor and a co-learner. In the inner loop, the feature extractor and the meta-learner update the parameters $\psi_i'$ and $\theta_i'$ with $M$-updates from a batch of $\mathcal{D}_i^S$, respectively. However, the co-learner does not update the parameters $\phi_i'$ in the inner loop. Therefore, task-specific parameters $\psi_i', \theta_i'$ and $\phi_i'$ are

---

**Algorithm 1** Cooperative Meta Learning

1: **[Meta-training]**
2: **Input**: Task distribution $p(\mathcal{T})$; Meta-learner model $f^m$; Co-learner model $f^c$; Step sizes $\alpha, \beta$; Loss scaling factor $\gamma$ ; The number of task in batch: N
3: **Output**: Meta-initialization parameters $\psi, \theta, \phi$
4: Randomly initialize parameters $\psi, \theta, \phi$
5: **while** not converged **do**
6:     Sample N tasks for batch $\mathcal{T}_i \sim p(\mathcal{T})$
7:     **for all** $\mathcal{T}_i$ **do**
8:         Sample dataset $\mathcal{D} = (\mathcal{D}_i^S, \mathcal{D}_i^Q)$ from $\mathcal{T}_i$
9:         Update task-specific parameters $(\psi_i', \theta_i')$ by Eq.(5)
10:     **end for**
11:     Intervene co-learner $\phi$ in meta-optimization step
12:     Calculate total loss with co-learner by Eq.(6)
13:     Update meta-initialization parameters $(\psi, \theta, \phi)$ with $\beta$ by Eq.(7)
14: **end while**
15: **return** $\psi, \theta, \phi$
16: **[Meta-testing]**
17: **Input**: Sample test dataset $\mathcal{D}_{test} = (\mathcal{D}_{test}^S, \mathcal{D}_{test}^Q)$
18: Load meta-initialization parameters $\psi, \theta, \phi$
19: **for all** $\mathcal{D}_{test}$ **do**
20:     **if** method is "CML" **then**
21:         Update task-specific parameters $(\psi', \theta')$ for $\mathcal{D}_{test}^S$ by Eq.(8)
22:         Evaluate the model $f^m_{\psi', \theta'}$ with $\mathcal{D}_{test}^Q$
23:     **end if**
24:     **if** method is "CML$^\dagger$" **then**
25:         Update task-specific parameters $\psi'$ for $\mathcal{D}_{test}^S$ by Eq.(9)
26:         Evaluate the model $f^c_{\psi', \phi}$ with $\mathcal{D}_{test}^Q$
27:     **end if**
28: **end for**

---

as follows:

$$(\psi_i', \theta_i') \leftarrow (\psi, \theta) - \alpha\nabla_{(\psi,\theta)}\mathcal{L}(f^m_{(\psi,\theta)}; \mathcal{D}_i^S), \quad \phi_i' = \phi \quad (5)$$

where $\alpha$ is an inner step size which is a fixed hyper parameters. Unlike $\psi_i'$ and $\theta_i'$ which are updated for the current task $\mathcal{T}_i$ in the inner loop, $\phi_i'$ keeps the updated parameters from the previously sampled tasks in the outer loop. In the outer loop, our model $f$ updates meta-initialization parameters from $\mathcal{D}_i^Q$ with task-specific parameters updated by $\mathcal{D}_i^S$. Our purpose is to converge to $\psi, \theta$ and $\phi$ that minimize Eq.(6) with the loss of the meta-learner and co-learner.

$$\sum_i^N \{\mathcal{L}(f_{(\psi_i', \theta_i', \phi_i)}; \mathcal{D}_i^Q)\} =$$
$$\sum_i^N \{\mathcal{L}(f^m_{(\psi_i', \theta_i')}; \mathcal{D}_i^Q) + \gamma\mathcal{L}(f^c_{(\psi_i', \phi)}; \mathcal{D}_i^Q)\} \quad (6)$$

where $\gamma$ is the loss scaling factor. The feature extractor and meta-learner have task-specific parameters $\psi_i'$ and $\theta_i'$

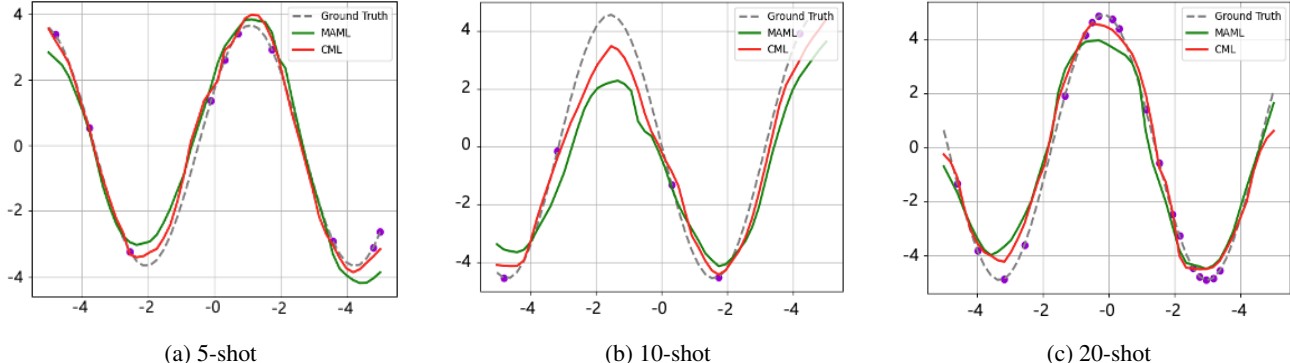

Figure 2: Results of MAML and CML on 5,10 and 20-shot of simple regression task.

with knowledge about the current task $\mathcal{T}_i$, but the co-learner has the parameters $\phi$ that have been updated by meta-optimization on the previous sampled N tasks.

$$(\psi, \theta, \phi) \leftarrow (\psi, \theta, \phi) - \beta \nabla_{(\psi, \theta, \phi)} \sum_i^N \{\mathcal{L}(f_{(\psi'_i, \theta'_i, \phi_i)}; \mathcal{D}_i^Q)\} \tag{7}$$

Then, we compute the meta-gradient with $N$ task losses for the query set $\mathcal{D}_i^Q$. It is created by gradient augmentation, where the gradient noise from the co-learner is added to the existing gradient. From Eq.(7), we update the meta-initialization parameters with an outer step size of $\beta$ by meta-optimization in the outer loop. The updated $\psi, \theta$ and $\phi$ are initialized with meta-initialization parameters for meta-testing. Lastly, our framework can infer with CML and CML$^\dagger$ using meta-testing dataset $D_{test} = (D_{test}^S, D_{test}^Q)$ in meta-testing phase, and CML performs the task-adaptation as follows:

$$(\psi', \theta') \leftarrow (\psi, \theta) - \alpha \nabla_{(\psi, \theta)} \mathcal{L}(f_{(\psi, \theta)}^m; \mathcal{D}_{test}^S) \tag{8}$$

The model then evaluates against $\mathcal{D}_{test}^Q$ by using the adapted parameters $\psi'$ and $\theta'$ like standard MAML. Therefore, it does not require any additional inference cost for the co-learner.

$$\psi' \leftarrow \psi - \alpha \nabla_{(\psi, \phi)} \mathcal{L}(f_{(\psi, \phi)}^c; \mathcal{D}_{test}^S) \tag{9}$$

On the other hand, in Eq.(9), CML$^\dagger$ performs the task-adaptation only for $\psi$. Note that $\phi$ of the co-learner does not perform task-adaptation and has existing meta-initialization parameters. Then we evaluate model with $\psi'$ and $\phi$ against $\mathcal{D}_{test}^Q$. Our CML algorithm is shown in Algorithm 1.

Next, we demonstrate that the gradient calculated from the co-learner converges theoretically when it is combined into meta-gradients. Our meta-gradient is updated by combining the gradients from the meta-learner($\theta'$) and co-learner($\phi$) in the feature extractor($\psi'$). We represent the loss function of the base network with $\psi'$, $\theta'$ to be $\mathcal{L}(\psi', \theta')$ after the task-adaptation.

Let the meta-initialization parameters of the base network consisting of $N$ feature extraction layers and the meta-learner as $\omega = \{\psi'_1, \cdots, \psi'_N, \theta'\}$. Consider the gradient $G^{(\psi', \theta')} = \{g_1^{\psi'}, \cdots, g_N^{\psi'}, g^{\theta'}\}$ of the base network computed by the meta-learner in the outer loop and the gradient $\bar{G}^{(\psi', \phi)} = \{\bar{g}_1^{\psi'}, \cdots, \bar{g}_N^{\psi'}, 0\}$ of the feature extractor computed by the co-learner. A zero value is just for matching the dimension. Let $\hat{G}^{(\psi', \theta', \phi)} = G^{(\psi', \theta')} + \bar{G}^{(\psi', \phi)} = \{(g_1^{\psi'} + \bar{g}_1^{\psi'}), \cdots, (g_N^{\psi'} + \bar{g}_N^{\psi'}), g^{\theta'}\}$ be the gradient of base network by gradient aggregation computed by the loss function $\mathcal{L}(\psi', \theta'; D^Q)$. If $\langle g_j^{\psi'}, \bar{g}_j^{\psi'} \rangle > 0, \; \forall j, (1 \leq j \leq N)$ is satisfied, the direction of the augmented gradient is a descent direction for finding meta-initialization parameters.

By Taylor's expansion of the loss function $\mathcal{L}$ for task and the base network of $\omega$ with CML updates:

$$\mathcal{L}(\omega - \alpha \hat{G}^{(\psi', \theta', \phi)}) = \mathcal{L}(\omega) - \alpha \nabla \mathcal{L}(\omega)^T \hat{G}^{(\psi', \theta', \phi)} + \mathcal{O}(\alpha^2)$$

With $\nabla \mathcal{L}(\omega) = G^{(\psi', \theta')}$ and $\lim_{\alpha \to 0} \frac{|\mathcal{O}(\alpha^2)|}{\alpha} = 0$, there exists $\bar{\alpha} > 0$ such that

$$\frac{|\mathcal{O}(\alpha^2)|}{\alpha} < |\langle G^{(\psi', \theta')}, \hat{G}^{(\psi', \theta', \phi)} \rangle|, \quad \forall \alpha \in (0, \bar{\alpha})$$

we have $|\langle G^{(\psi', \theta')}, \hat{G}^{(\psi', \theta', \phi)} \rangle| > 0$ ($\because \langle g_j^{\psi'}, \bar{g}_j^{\psi'} \rangle > 0, \; \forall j$). In this condition, $\mathcal{L}(\omega - \alpha \hat{G}^{(\psi', \theta', \phi)}) - \mathcal{L}(\omega) < 0$ and $\forall \alpha \in (0, \bar{\alpha})$. Therefore, CML updates the parameters $\omega$ toward the descent direction in the outer loop.

## 4 EXPERIMENTS

In this section, we apply our CML to various gradient-based meta-learning and evaluate the performance of our framework on few-shot regression, few-shot image classification and few-shot node classification in Section 4.1~4.3. We also conduct a gradient analysis of the co-learner in our framework, as discussed in Section 4.4. To confirm the performance improvement from the gradient augmentation

effect in CML, not from additional parameters or multi-branch structure, we compare CML with CL, having the same structure, and the naive gradient-based meta-learning in Section 4.5. To a fair comparison, we follow the original settings of several gradient-based meta-learning algorithms and test them on well-known few-shot benchmarks. More implementation details are in the Appendix.

## 4.1 FEW-SHOT REGRESSION

We evaluate the performance of CML with MAML as a baseline in K-shot sinusoidal regression. The amplitude and phase of the sinusoidal wave follow the ranges of [0.1,5.0] and $[0,\pi]$. Each task consists of datapoints $\mathbf{x}$, $\mathbf{y}$ of a sinusoidal wave. The input $\mathbf{x}$ is uniformly sampled in the range [-5.0,5.0]. The loss function for comparing predicted $\mathbf{y}$ and target $\mathbf{y}$ uses mean-squared error. The baseline consists of 2 hidden layers of size 40 with ReLU nonlinearities, 1 input layer and 1 output layer following [Finn et al., 2017]. For CML, the regressor is additionally attached with 1 hidden layer of size 40 with ReLU nonlinearities and 1 output layer as a co-learner. In meta-training, we use K $\in \{5,10,20\}$ samples as training examples and train using a batch size of 4, one inner-gradient step, a fixed step size of 0.01 and our loss scaling factor $\gamma$ of 0.2. For meta-testing, we evaluate adaptation with one gradient step for K=5, 10, and 20 test points. Each model predicts the target sinusoidal wave through the given K test points. Furthermore, the co-learner is deleted and it is only evaluated from the feature extractor and meta-learner as the original model. Figure 2 shows that our CML performs better than MAML for 5, 10, and 20 shots. It means that our framework adapts well to simple networks and shows better generalization performance than the original framework.

## 4.2 FEW-SHOT IMAGE CLASSIFICATION

We compare the performance of the proposed method on few-shot image classification with several gradient-based meta-learning algorithms including MAML [Finn et al., 2017], MAML++ [Antoniou et al., 2018], BOIL [Oh et al., 2020] and Sharp-MAML [Abbas et al., 2022]. In this experiment, we evaluate the performance of 5-way 1/5-shot problems on MiniImagenet datasets. In CML, the co-learner uses two convolution layers and a fully connected layer. Our loss scaling factor $\gamma$ is fixed at 0.5 for all methods. We also evaluate the performance of the co-learner.

**Results** Table 1 shows that the proposed methods outperform the original algorithms. Note that CML, which removed the co-learner during meta-testing, improves the performance of the original algorithms. It indicates that the co-learner only performs meta-optimization, which successfully leads it to converge to well-generalized meta-initialization parameters. Specifically, on MAML++ [Anto-

Table 1: Test accuracy of 4-conv network with the CML framework on MiniImagenet dataset. The MAML algorithms are from [Oh et al., 2020]. The Sharp-MAML is used for reproduction. The blue color and red color indicate the output of the meta-learner and co-learner, respectively. Our experiments are performed in 3 runs.

| Method | MiniImagenet 5-way (%) | |
| --- | --- | --- |
| | 1-shot | 5-shot |
| MAML [Finn et al., 2017] | $47.44 \pm 0.23$ | $61.75 \pm 0.42$ |
| MAML + CML | $49.32 \pm 0.37$ | $65.84 \pm 0.46$ |
| MAML + CML$^\dagger$ | $50.35 \pm 0.15$ | $66.43 \pm 0.43$ |
| MAML++ [Antoniou et al., 2018] | $52.15 \pm 0.26$ | $68.32 \pm 0.44$ |
| MAML++ + CML | $52.46 \pm 0.05$ | $70.08 \pm 0.61$ |
| MAML++ + CML$^\dagger$ | $52.86 \pm 0.17$ | $70.69 \pm 0.49$ |
| BOIL [Oh et al., 2020] | $49.61 \pm 0.61$ | $66.45 \pm 0.37$ |
| BOIL + CML | $50.04 \pm 0.30$ | $66.91 \pm 0.13$ |
| BOIL + CML$^\dagger$ | $50.83 \pm 0.25$ | $67.50 \pm 0.48$ |
| Sharp-MAML [Abbas et al., 2022] | $49.06 \pm 0.52$ | $65.63 \pm 0.54$ |
| Sharp-MAML + CML | $49.56 \pm 0.45$ | $66.90 \pm 0.20$ |
| Sharp-MAML + CML$^\dagger$ | $49.70 \pm 0.62$ | $67.06 \pm 0.16$ |

Table 2: Test accuracy for 5-way 1/5-shot of the MAML and CML framework on the diverse datasets.

| Method | Omniglot (%) | | CIFAR-FS (%) | | FC100 (%) | | VGG Flower (%) | |
| --- | --- | --- | --- | --- | --- | --- | --- | --- |
| | 1-shot | 5-shot | 1-shot | 5-shot | 1-shot | 5-shot | 1-shot | 5-shot |
| MAML | 91.78 | 96.59 | 56.55 | 70.10 | 36.07 | 48.03 | 63.17 | 74.48 |
| CML | **93.99** | **97.15** | **57.67** | **73.87** | **36.90** | **51.06** | **64.31** | **77.03** |

niou et al., 2018], our framework achieves 70.08% performance without any additional inference cost in meta-testing. In CML$^\dagger$, we infer through the co-learner instead of the meta-learner. CML$^\dagger$ outperforms CML because CML$^\dagger$ has more parameters. More interestingly, the co-learner shows high performance without any adaptation. This suggests that our framework has a well-trained feature extractor, and the co-learner plays an important role in achieving this. It looks similar to BOIL [Oh et al., 2020], but whereas BOIL freezes the meta-learner for representation changes, we introduce a co-learner to take advantage of the gradient augmentation effect of Theorem 3.2. In other words, the co-learner provides a gradient augmentation effect to converge the feature extractor with meta-initialization parameters that enable good generalization. The effectiveness of this approach is also demonstrated across different datasets, as shown in Table 9.

## 4.3 FEW-SHOT NODE CLASSIFICATION

In this experiment, we evaluate CML on a few-shot node classification of graph neural networks (GNNs). Few-shot node classification aims to achieve fast adaptation to new node tasks that are unseen during training. We also define an N-way K-shot problem and consider node tasks $\mathcal{T}_{node}$ which consist of support nodes $\mathcal{D}^{\mathcal{S}}_{node}$ and query nodes $\mathcal{D}^{\mathcal{Q}}_{node}$. For performance comparison, we use G-Meta [Huang and Zitnik, 2020] and AMM-GNN [Wang et al., 2020], which belong to

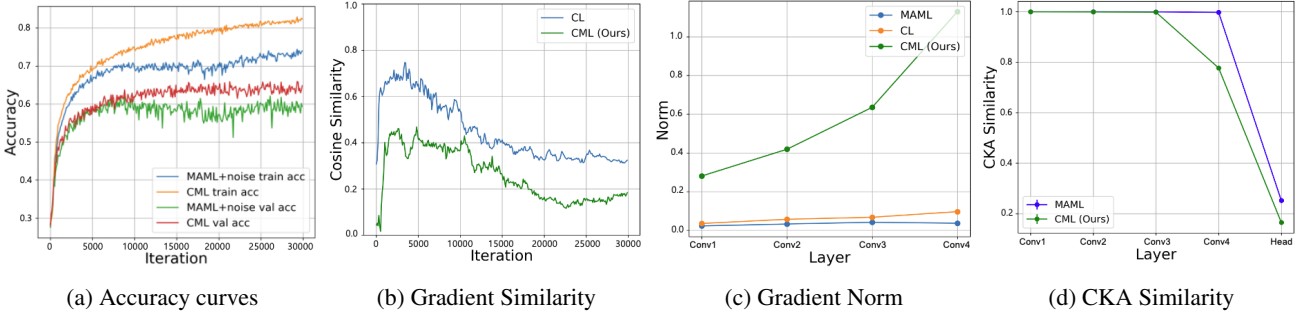

|  (a) Accuracy curves | (b) Gradient Similarity | (c) Gradient Norm | (d) CKA Similarity |

Figure 3: **(a)** Accuracy of MAML with random noise and CML. **(b)** Gradient similarity for the meta-learner and co-learner of the 4th convolution layer. **(c)** Comparison of gradient norm for the feature extractor in MAML, CL and CML after task-adaptation in the inner loop. At this point, we ignore the effect of bias, because of its negligible impact. **(d)** CKA Similarity results of representations before and after task-adaptation in the inner loop.

Table 3: Results on node classification with CML. The G-Meta and AMM-GNN algorithms are from [Tan et al., 2022]. Our all experiments are performed 5 runs.

| Method | CiteSeer 2-way (%) | | Amazon 2-way (%) | | CoraFull 5-way (%) | |
|---|---|---|---|---|---|---|
| | 1-shot | 5-shot | 1-shot | 5-shot | 1-shot | 5-shot |
| G-Meta | 55.15 | 64.53 | 70.57 | 85.96 | 60.44 | 75.84 |
| G-Meta + CML | **61.17** | **76.07** | **72.26** | **87.10** | **60.49** | **76.02** |
| AMM-GNN | 54.53 | 62.93 | 74.29 | 80.10 | 58.77 | 75.61 |
| AMM-GNN + CML | **61.13** | **66.88** | **78.91** | **86.68** | **63.27** | **76.19** |

gradient-based meta learning with GNN, as a baseline and evaluate on the CoraFull, Amazon-Computer and CiteSeer datasets [Sen et al., 2008, Shchur et al., 2018]. We also perform 5-way 1/5-shot and 2-way 1/5-shot, respectively. Our base model follows [Tan et al., 2022], using GCN as an encoder of hidden size 16 and a fully connected layer as a meta-learner. We train using the Adam optimizer for a step size of 0.001. Also, we set the inner gradient-steps of 20 with a step size of 0.05. In CML framework, our co-learner additionally includes 1 hidden layer of size 16 and a fully connected layer as the output layer using the loss scaling factor $\gamma$ of 0.2. As shown in Table 3, our framework outperforms the baseline method on node classification. In this experiment, the co-learner improves the performance of the encoder and meta-learner, despite having a very simple network structure. It shows that our framework is suitable for solving the few-shot problems and is applicable to various DNN methods related to meta-learning.

## 4.4 GRADIENT AUGMENTATION ANALYSIS

In this section, we analyze the effect of gradient augmentation by a co-learner. All of our experiments are evaluated on MiniImagenet 5-way 5-shot, and the network structure and experimental settings are as in Section 4.2.

**Is the gradient of the co-learner really meaningful?** To verify that the gradient of the co-learner is applied as mean-

ingful noise on the meta-gradient, we compare it to MAML with random noise. To generate random noise, we introduce a co-learner that does not perform any updating into the MAML (e.g. inner and outer loops). By doing so, the meta-gradient of MAML is updated with a randomized gradient added to the original gradient. For a fair comparison, both models have the same initialization parameters and take the same sampled data as input. Our CML outperforms MAML with random noise and converges to well-generalized parameters much faster, as shown in Figure 3a. It shows that the gradient of the co-learner influences the meta-gradient with meaningful noise, not just random noise.

**The co-learner induces the diversity of the meta-gradient** Yang et al. [2020] learns a well-generalized full network by inducing gradient diversity with multiple-forwarding of subnetworks. Inspired by this, we perform gradient augmentation on the meta-gradient by updating the proposed co-learner only in the outer loop, unlike the meta-learner, to induce gradient diversity. To demonstrate this, we compare it to Collaborative Learning (CL) [Song and Chai, 2018], a multi-branch framework approach that does not freeze the co-learner, i.e., CL is like a multi-head framework that updates both the meta-learner and the co-learner. Oh et al. [2020] shows that the convolution layer before the classifier on task-adaptation is the key to inducing representation change. Based on their findings, we compare CL and CML by computing the gradient similarity of the 4th convolution layer of the feature extractor calculated from each meta-learner and co-learner. From Figure 3b, we observe that CML has a lower gradient similarity between the meta-learner and co-learner in the feature extractor than CL. In general, a value closer to 1 suggests that the patterns and features captured are more similar. Our CML explores more different directions during the optimization process than CL due to its co-learner. It indicates that the co-learner produces a notably more diverse gradient, attributed to the augmentation effect within the meta-gradient. We also show that the gradient similarity of CML is larger than zero, which

Table 4: Number of parameters and test accuracy on Mini-Imagenet 5-way 1/5-shot. CML and CL use the MAML framework as a baseline. The "⋆" and "†" indicate the output of the meta-learner and co-learner, respectively. All experiments are performed in 3 runs.

| Method | Parameters # | | MiniImagenet 5-way (%) | |
|---|---|---|---|---|
| | Train | Test | 1-shot | 5-shot |
| MAML | 129K | 129K | $47.44 \pm 0.23$ | $61.75 \pm 0.42$ |
| More-MAML | 232K | 232K | $48.48 \pm 0.60$ | $62.53 \pm 0.12$ |
| CL | 203K | 203K | $47.57 \pm 0.15$ | $62.36 \pm 1.02$ |
| *(1) Comparison to the meta-learner* | | | | |
| CL⋆ | 203K | 129K | $47.45 \pm 0.13$ | $61.60 \pm 1.39$ |
| CML (Ours) | 203K | 129K | $\mathbf{49.32 \pm 0.37}$ | $\mathbf{65.84 \pm 0.46}$ |
| *(2) Comparison to the co-learner* | | | | |
| CL† | 203K | 195K | $48.45 \pm 0.40$ | $62.50 \pm 0.62$ |
| CL† w/o adaptation | 203K | 195K | $20.66 \pm 0.38$ | $20.54 \pm 2.61$ |
| CML† (Ours) | 203K | 195K | $\mathbf{50.35 \pm 0.15}$ | $\mathbf{66.43 \pm 0.43}$ |

Table 5: Ablation study of the loss scaling factor.

| Loss scaling factor ($\gamma$) | MiniImagenet 5-way (%) | |
|---|---|---|
| | 1-shot | 5-shot |
| 0.2 | 49.07 | 64.39 |
| 0.5 | **49.61** | 65.53 |
| 0.8 | 49.07 | 64.73 |
| 1.0 | 49.32 | **65.84** |

satisfies the precondition in Theorem 3.2.

**Effect of the gradient augmentation** In this experiment, we investigate the impact of the augmentation on the meta-gradient. Firstly, we analyze the norm of the gradient for each convolution layer in the feature extractor after task-adaptation in the inner loop. The gradient norm is an important indicator of how much a particular layer affects learning. Figure 3c shows the averaged gradient norm of each convolution layer in the feature extractor for MAML, CL and CML. We observe that CL and MAML have very small gradient norms, close to zero on all convolution layers. It indicates that they mostly maintain the existing representation with minimal changes for a new task. However, our framework has relatively larger gradient norms, which indicates that the model is dynamically adapting to new tasks, and there is an amplification of diversity on the meta-gradient from the co-learner. We also perform a Centered Kernel Alignment (CKA) [Kornblith et al., 2019] to compare representations similarity before and after adaptation. CKA is one way to compare the similarity of representation and a CKA value close to 1 means that the two representations are similar. Figure 3d shows the CKA similarity of MAML and CML representations before and after task-adaptation. In MAML, the change in representation occurs only at the head. On the other hand, CML indicates that the representation change occurs in the 4th convolution layer, which also proves that the co-learner in CML induces the representation change at a high level. Thus, our results suggest that a new meta-gradient from the co-learner induces it to learn more task-specific features.

## 4.5 EFFICIENCY ANALYSIS OF THE CML STRUCTURE

In this section, we conduct an experiment to justify the validity of our framework's structure. Our framework requires more parameters during meta-training due to the addition of the co-learner. Therefore, we compare the parameter sizes of CML, CL and MAML with more parameters, called More-MAML, to demonstrate that our framework does not simply improve performance by having more parameters. In this experiment, CML and CL follow the same network architecture as Section 4.2, while More-MAML has additional convolution layers. From Table 4, we can see that More-MAML, CL and CML have 232K, 203K and 203K parameters, respectively, during meta-training. Note that the CML has fewer parameter sizes than More-MAML and CL in meta-testing, but shows better performance on Mini-Imagenet datasets. Also in (1), CL⋆ shows a performance degradation when inferring with a meta-learner like CML. In setting (2), both CL† and CML† use the co-learner to evaluate performance. We observe that without performing adaptation, CL† leads to a deterioration in the model's inferential capabilities. It emphasizes that adaptation is essential in the general case, and that our approach has a uniquely structured framework. Notably, although our co-learner does not perform task-adaptation during meta-testing in CML†, it outperforms models with a similar number of parameters while achieving the highest accuracy. In this experiment, our findings highlight that having more parameters in meta-learning does not necessarily lead to improved performance, while our framework demonstrates an effective learning framework to address this limitation.

## 4.6 ABLATION STUDY

**Update scheme for loss scaling factor** The proposed method has parameters $\gamma$ for the influence of the co-learner on the feature extractor. To verify the effect of this influence, we conduct experiments for 5-way 1/5-shot on MiniImagenet datasets. From Table 5, we show that our method has higher performance than conventional MAML regardless of $\gamma$. In particular, 1 shot and 5 shot achieve the highest performance at 0.5 and 1.0, respectively. This result shows that our method is robust against $\gamma$ and suggests that the intervention of the co-learner is important.

## 5 CONCLUSION AND DISCUSSION

In this paper, we propose a novel training framework called Cooperative Meta-Learning (CML). The main idea of our framework is that the proposed co-learner in meta-training generates a gradient augmentation effect. To achieve this, we

design the co-learner so that it only updates in the outer loop and can be easily deleted in meta-testing. Our experiments demonstrate that our co-learner generates meaningful gradients, which leads to diversity on the meta-gradient, and this guides the learning direction to better meta-initialization parameters. It also shows that the diversity of the meta-gradient is a key factor in its strong generalization ability in the few-shot problem.

## Acknowledgements

This work was carried out with the support of "Cooperative Research Program for Agriculture Science and Technology Development (Project No. RS-2024-00332198 )" Rural Development Administration, Republic of Korea.

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

# Cooperative Meta-Learning with Gradient Augmentation
## (Supplementary Material)

**Jongyun Shin**[1]                **Seungjin Han**[1]                **Jangho Kim**[*1]

[1]Computer Science Department, Kookmin University, Seoul, Korea
`{whddbs519,gkstmdwls99,jangho.kim}@kookmin.ac.kr`

# A   IMPLEMENTATION DETAILS

## A.1   IMAGE CLASSIFICATION

In our experiments, BOIL [Oh et al., 2020] and MAML++ [Antoniou et al., 2018] demonstrate results that are highly consistent with the original papers, thus reporting the original paper results [1] [2]. In addition, the MAML [Finn et al., 2017] follows the experiments in [Oh et al., 2020]. However, in the case of Sharp-MAML, the results obtained using the official code in the same experimental setup differed from the original paper. Therefore, we report the experimental results based on our execution following the official code [3].

**Architecture** we used the 4-conv network model, following [Finn et al., 2017]. In detail, the model contains four $3 \times 3$ convolution layers with batch normalization, a ReLU nonLinearity and $2 \times 2$ max-pooling and a fully connected layer. CML additionally includes two $3 \times 3$ convolution layers and a fully connected layer as a co-learner.

**Experimental settings** We basically follow the original settings for each algorithm. For task-adaption, We follow the original settings: 5 inner-gradient steps on Sharp-MAML and MAML++ and 1 inner-gradient step on the rest following [Oh et al., 2020]. In CML framework, we train using loss scaling of $\gamma = 1$. We perform 3 runs and report all our results from the model with the best validation accuracy. We used the Pytorch framework and GeForce RTX 3090 for all experiments.

**Datasets** We evaluate our method on the following benchmark datasets. **MiniImagenet** contains 60000 images with 100 classes and 600 images size of $84 \times 84$ for each class. **Omniglot** contains 32,460 images size of $28 \times 28$ of handwritten characters with 1,623 different characters from 50 alphabets. **CIFAR-FS** is randomly sampled based on CIFAR-100 and it contains 600 images size of $32 \times 32$ with 100 classes. **FC-100** is also a split dataset from CIFAR-100 that contains 600 images size of $32 \times 32$ with 100 classes. **VGG-Flower** contains 258 images size of $32 \times 32$ for each class.

Table 6: Statistics datasets

| Dataset | Nodes # | Edge # | Features # | Class split (train / validation / test) |
|---|---|---|---|---|
| CoraFull | 19,793 | 63,421 | 8,710 | 40 / 15 / 15 |
| Amazon-Computer | 13,752 | 245,861 | 767 | 4 / 3 / 3 |
| CiteSeer | 3,327 | 4,552 | 3,703 | 20 / 10 / 10 |

---

[1]`https://github.com/jhoon-oh/BOIL`

[2]`https://github.com/AntreasAntoniou/HowToTrainYourMAMLPytorch`

[3]`https://github.com/mominabbass/Sharp-MAML`

*Accepted for the 40th Conference on Uncertainty in Artificial Intelligence* (UAI 2024).

## A.2 NODE CLASSIFICATION

We perform our experiments with the same environment from the official code [4] of [Tan et al., 2022]. We experiment on the CoraFull, Amazon-Computer and CiteSeer datasets from Table 6, which are widely used in node classification.

## B COMPARISON OF THE META-LEARNER AND CO-LEARNER FOR THE SAME PARAMETERS

Table 7: Results on the performance of the meta-learner and co-learner with the same parameters. Our CML framework uses MAML as a baseline with a shared feature extractor, meta-learner, and co-learner. The "$*$" indicates that the model failed to converge.

| Structure | Learner | MiniImagenet 5-way (%) | |
|---|---|---|---|
| | | 1-shot | 5-shot |
| Conv(0) | Meta-learner | $\mathbf{49.52 \pm 0.41}$ | $\mathbf{65.82 \pm 0.55}$ |
| | Co-learner | $49.16 \pm 0.47$ | $65.13 \pm 0.27$ |
| Conv(2) | Meta-learner | $\mathbf{48.06 \pm 0.97}$ | $\mathbf{66.20 \pm 0.44}$ |
| | Co-learner | $*$ | $65.44 \pm 0.27$ |

We conduct experiments on the performance of the meta-learner and co-learner with the same capacity. In this experiment, our feature extractor has the four convolution layers as shown in Section 4.2, and the co-learner and meta-learner are evaluated on the same structure, Conv(0) with no convolution layer and only a fully connected layer, and Conv(2) with two convolution layers and a fully connected layer. From Table 7, we observe that the meta-learner that performs task-adaptation in meta-testing achieves higher performance than the co-learner. It can be seen that our feature extractor already has good performance during meta-training, rather than the co-learner having good performance despite not performing task-adaptation. Therefore, the co-learner assists the learning of the feature extractor during meta-training and guides it to converge in a good direction. However, we find that our co-learner fails to converge on the 1-shot problem with the Conv(2) structure. This suggests that we need to empirically evaluate the optimized structure of the co-learner based on the network architecture.

## C ABLATION STUDY ON THE NUMBER OF CONV-LAYER IN THE CO-LEARNER

Table 8: Test Accuracy(%) by number of convolution layer on MiniImagenet 5-way 1-shot. We use MAML as a baseline and follow the experimental settings in Section 4.2.

| Conv layer (#) | Conv(0) | Conv(1) | Conv(2) | Conv(3) | Conv(4) |
|---|---|---|---|---|---|
| CML | $\mathbf{49.52 \pm 0.41}$ | $49.39 \pm 0.51$ | $49.32 \pm 0.37$ | $48.85 \pm 0.28$ | $49.36 \pm 0.23$ |
| CML$^\dagger$ | $49.16 \pm 0.47$ | $50.21 \pm 0.60$ | $\mathbf{50.35 \pm 0.15}$ | $49.82 \pm 0.29$ | $50.17 \pm 0.28$ |

We explore the impact for the structure of the co-learner in our CML framework. Table 8 shows that all models outperform the performance of standard MAML. In particular, Conv(0), which has only a fully connected layer without a convolution layer, achieved the highest performance. It clearly shows that our learning framework is effective in leading convergence to well-generalized meta-initialization parameters. Also, the co-learner in the Conv(2) model with two convolution layers achieves the highest accuracy of 50.35%.

## D CML TO A LARGER NETWORK

We run experiments on CIFAR-FS, VGG-Flower, and FC-100 on a larger network. Resnet12 [Oreshkin et al., 2018] network.The findings demonstrate that CML enhances the performance of MAML, even with larger backbone architectures. This improvement can be attributed to the enhanced representational ability of the feature extractor facilitated by the co-learner, irrespective of the backbone network size.

---

[4] https://github.com/Zhen-Tan-dmml/TLP-FSNC

Table 9: Test accuracy of Resnet12 network with the CML framework on CIFAR-FS, VGG Flower, and FC100 dataset.

| Method | CIFAR-FS (%) | | VGG Flower (%) | | FC100 (%) | |
|--------|--------|--------|--------|--------|--------|--------|
| | 1-shot | 5-shot | 1-shot | 5-shot | 1-shot | 5-shot |
| MAML | 61.86 | 73.32 | 63.43 | 75.42 | 36.61 | 47.48 |
| CML | **62.11** | **78.31** | **66.07** | **81.15** | **37.56** | **51.51** |

# E  VISUALIZATION OF CML BY T-SNE

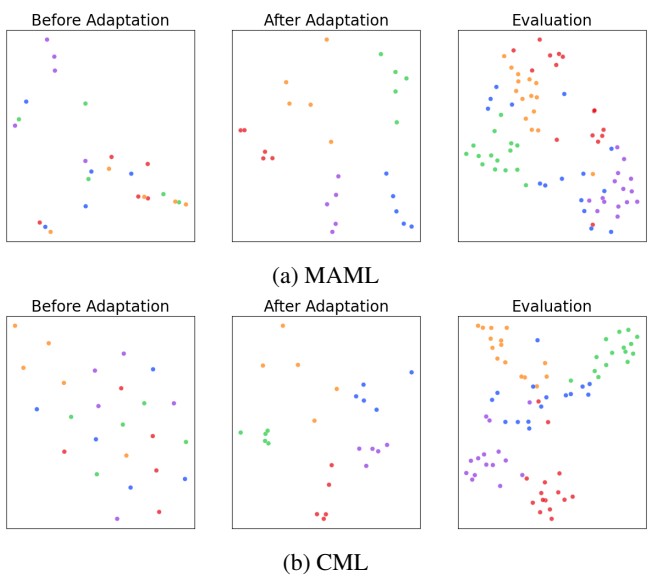

(a) MAML

(b) CML

Figure 4: t-SNE of (a) MAML and (b) CML on trained miniimagenet. We perform the adaptation with the support set and then evaluate the method with the query set.

T-SNE [Van der Maaten and Hinton, 2008] is a typical dimension reduction technique that maps high-dimensional data into a lower-dimensional space. It allows us to assess the similarity of data points before and after adaptation. We experiment with T-SNE for MAML and CML without the co-learner with the same parameters at inference time. In Table 4, our CML shows that the ability to form more consistent and distinct clusters than MAML. It can be seen that the intervention of the co-learner attached to the CML produces a gradient augmentation effect, which is attributed to better generalization performance.

