# OpenReview forum: "Cooperative Meta-Learning with Gradient Augmentation"
_auai.org/UAI/2024/Conference — UAI 2024 poster_

### Official Review · Reviewer_Zo2J · 2024-03-14

**Q2-1 Originality-Novelty:** 2
**Q2-2 Correctness-Technical Quality:** 3
**Q2-5 Clarity Of Writing:** 3

**Q1 Summary And Contributions:**

The Model Agnostic Meta-Learning (MAML) algorithm [Finn et al., 2017] is designed to learn a model that can be fine-tuned to new tasks with limited data. This paper proposes "the co-learner," a novel classification head utilized exclusively during the outer loop of training, to update the weights of the feature extraction sub-network. Through a series of experiments, the paper demonstrates the contribution of this enhancement across several MAML variants, tasks, and datasets.

**Q2-3 Extent To Which Claims Are Supported By Evidence:**

3: Good: the main claims are supported by convincing evidence (in the form of adequate experimental evaluation, proofs, (pseudo-)code, references, assumptions).

**Q2-4 Reproducibility:**

2: Fair: key resources (e.g. proofs, code, data) are unavailable but key details (e.g. proof sketches, experimental setup) are sufficiently well-described for an expert to confidently reproduce the main results.

**Q3 Main Strengths:**

1.	The proposed enhancement is simple, with clear motivation, and a novel addition to the MAML framework
2.	The enhancement can be applied to multiple MAML variants
3.	The enhancement consistently leads to better results in the presented experiments

**Q4 Main Weakness:**

1.	The first part of the paper, which presents the core idea and its motivation, is easy to read and follow. However, the second part is more difficult to follow.  See detailed comments
2.	The suggested enhancement is very similar to the collaborative learning (CL) framework [Song and Chai, 2018] (page ).

**Q5 Detailed Comments To The Authors:**

1.	Why does Equation 9 show a gradient step w.r.t. to co-learner parameters $\phi$ if it does not update it?  Why not show a gradient w.r.t. to feature extractor $\psi$ only?
2.	Theorem 1 is given on page 5 without any context or motivation
3.	In Theorem 1: why is the gradient of the feature extractor computed by the co-learner identical to the one by the meta-learner, but with negative signs?? When looking at the proof, I could only understand the first line (Taylor’s expansion)
4.	In the last sentence on page 5, the period should be a comma.
5.	First sentence below Table 2: “CML$^{\dagger}$ outperforms CML because CML$^{\dagger}$  has more parameters” – how can you be certain of this statement?  Did you try increasing the number of parameters in CML to verify if indeed the performance becomes the same?
6.	Section 4.3: Please provide references for the CoraFull and citepSeer datasets.
7.	How was the scaler factor \gamma selected for the experiments?  What is its value in Section 4.2?
8.	Page 7 “CL is like a multi-head framework that updates both the meta-learner and the co-learner”: this sentence implies that CML is identical to CL with the exception that CML does not update the co-learner during the inner loop. Is this correct?
9.	Are the results in Section 4.4 obtained by using the datasets and networks in Section 4.2?  If yes – please state so clearly at the beginning of this section.

**Q9 Complying With Reviewing Instructions:**

Yes

---

> ### Author Rebuttal · Authors · 2024-04-07
>
> Thank you very much for Reviewer Zo2J meticulous and precise analysis of our work! We are delighted that our motivation for the straightforward methods we propose to enhance the MAML framework has been effectively communicated.
>
> **[Q1]**
> Collaborative Learning (CL) [1]’s design was proposed for a general classification problem, not a meta-learning task divided into inner loop and outer loop, and we apply it to meta-learning. Nevertheless, CL and we differ significantly in how we train with and without updates from the co-learner.
>
> Because CL requires the co-learner to always adapt to new tasks in the inner loop, the gradient of the co-learner must be computed for the feature extractor during meta-testing. If CL does not adapt the co-learner like CML, performance may suffer. In Table 4-(2), CL$^{\dagger}$ (w/o adaptation) shows this. However, our CML method can easily remove the co-learner during meta-testing because the gradient of the co-learner does not intervene in the inner loop, and it outperforms CL, which requires the co-learner all the time. Moreover, CML$^{\dagger}$ achieves good performance without updating the co-learner in the inner loop. Thus, we emphasize that CML's learning strategy is an effective meta-learning framework that utilizes the advantages of both the CL and MAML frameworks.
>
> **[Q2]**
> As you understand, we do not update co-learner in the inner loop and it is implemented in the code by setting 0 for the learning rate of co-learning in the inner loop in our supplementaries. Our intention is that for CML$^\dagger$, the parameters of both the co-learner and the feature extractor are utilized during meta-testing. Therefore, we emphasize that the updated feature extractor for tasks infers with the co-leaner.
>
> However, we believe that the notation of the expression can cause confusion about the co-learner's updates, so we will remove the notation of the co-learner to reflect that, as you mentioned.
>
> **[Q3]**
> We aim to see if the meta-gradient (Eq.6) converges theoretically when the gradient of the co-learner is combined with it of  the meta-learner in the feature extractor. As stated as
> “We demonstrate the convergence of the meta-gradient as a result of the co-learner’s gradient". Our meta-gradient is updated by combining the gradients from the meta-learner($\theta^{\prime}$ ) and co-learner($\phi$) in the feature extractor($\psi^{\prime}$ ). To make our goal explicit, we will revise the motivation of the theory in detail.
>
> **[Q4]**
> Thanks for the good question. As you mentioned, the gradient of the feature extractor computed by the co-learner is the same as that computed by the meta-learner. However, we split it to emphasize that the gradient of the feature extractor is the sum of the gradients of the co-learner and the meta-learner, resulting in a new gradient flow. Note that the use of 'bar' (=negative sign you mentioned) is intended to represent a different notation from that of the gradient notation used for the meta-learner, rather than indicating any association with negative/positive signs.
>
> During the actual computation in backpropagation, the gradient of the feature extractor is propagated by an augmented gradient that is the sum of the gradients of the co-learner and the meta-learner in the last N-th layer of the feature extractor, following the chain rule.
>
> To aid understanding, we have further organized the equations below for presentation. We will ensure clarity in the final version to avoid any confusion.
>
> Additional equations are continued in Part 2.
>
> (Unfinished)

---

### Official Review · Reviewer_HZvF · 2024-03-19

**Q2-1 Originality-Novelty:** 2
**Q2-2 Correctness-Technical Quality:** 3
**Q2-5 Clarity Of Writing:** 3

**Q1 Summary And Contributions:**

This paper proposes cooperative meta-learning with gradient augmentation by injecting noise into the gradient update of the meta-learning model through a co-learner. Extensive experiments on multiple datasets and competitive baselines demonstrate the effectiveness of the proposed method.

**Q2-3 Extent To Which Claims Are Supported By Evidence:**

3: Good: the main claims are supported by convincing evidence (in the form of adequate experimental evaluation, proofs, (pseudo-)code, references, assumptions).

**Q2-4 Reproducibility:**

3: Good: key resources (e.g. proofs, code, data) are available and key details (e.g. proofs, experimental setup) are sufficiently well-described for competent researchers to confidently reproduce the main results.

**Q3 Main Strengths:**

**Strength:**


* This paper is well-written and easy to follow.

* Experiments are extensive on multiple datasets.

**Q4 Main Weakness:**

**Weakness:**


* Injecting noise into gradient descent has been proposed in existing works[1] to improve the meta-learning generalization performance.  There is no discussion and comparisons in related works. This diminishes the novelty of this work.


* Since the proposed method needs a co-learner for training the model in the outer-loop, it would be better to provide the training efficiency compared to the base model, MAML.


* It is unclear why injecting noise into the gradient update of the meta-learning model can obtain better meta-initialization, could you provide more intuition and explanations?


* It would be better to provide more results with a larger backbone network.


Reference:


[1] Probabilistic Model-Agnostic Meta-Learning,  NeurIPS 2018

**Q5 Detailed Comments To The Authors:**

Please refer to the weakness section.

**Q9 Complying With Reviewing Instructions:**

Yes

---

> ### Author Rebuttal · Authors · 2024-04-07
>
> Thank you, Reviewer HZvF for reviewing our work! We sincerely thank you for your detailed and constructive feedback on the development of this paper.
>
> **[Q1]**
> Both [1] and our paper involve noise injection; however, our approach differs from theirs in that we specifically focus on gradient augmentation.
>
> Previously proposed methods were deterministic MAML that utilized past experience to learn a prior for a task, but due to the ambiguity and uncertainty of obtaining a single model in a few-shot learning problem, [1] proposes a probabilistic MAML. Therefore, they perform meta-training, assuming the model distribution is a Guassian distribution, and sample potential solutions from the model distribution in meta-testing.
> [1] introduces noise, denoted Vq, into the mean value during the gradient descent process of the outer loop to facilitate learning a model distribution with potential solutions. This noise injection occurs directly into the model distribution during the forward process, which is equivalent to adding noise weights to the model weights. However, the proposed CML does not approximate the model as a distribution, and the noise generated by the co-learner directly affects the original gradient.
>
> Their method specifically focuses on scenarios involving ambiguity in class or function sampling. They also do not have experiments on relatively less ambiguous few-shot image classification in their paper. We tried reimplementing according to their unofficial code [2] and failed to converge. Based on our method, we achieve 65.84 for MiniImagenet 5-shot and 61.85 for MAML, while their method fails to converge around 46.21. The issue in [2] also had an problem with reimplementing of training, and we have a similar issue.
>
> Therefore, we think that our proposed method is different from [1] in its purpose, motivation and method for injecting noise.
>
> **[Q2]**
> We conduct experiments to compare the training efficiency of CML and MAML, and all experiments follow Section 4.2.
>
> The proposed method requires additional computational cost due to the co-learner during meta-training. As a result, we find that the training time increases by about 14% compared to MAML.
>
> | Method | Training Time (Hour) | Accuracy |
> |:------:|:--------------------:|:--------:|
> |  MAML  |         1.1H         |   61.75  |
> |   CML  |         1.2H         |   65.84  |
>
> However, our supplementary experiments show that the model exhibits fast convergence once the co-learner is introduced in the outer loop. To illustrate this, we provide validation accuracy results for each epoch. Our findings show that CML converges faster than MAML and actually requires less training time to achieve equivalent accuracy.
>
> | Epochs |   1   |   10  |   60  |  100  |  200  |  300  |
> |:------:|:-----:|:-----:|:-----:|:-----:|:-----:|:-----:|
> |  MAML  | 22.79 | 31.47 |  56.2 | 58.02 | 61.71 | 62.27 |
> |   CML  | 25.81 | 39.76 | 58.92 | 60.88 | 63.95 | 65.21 |
>
> For example, let's compare the time to achieve MAML's best validation accuracy. The time taken to achieve the best accuracy of MAML is 0.65H, and the training time taken by CML to achieve this accuracy is 0.47H. This means that our CML actually takes less time to achieve the same accuracy with a faster convergence rate.
>
> In conclusion, our experiments show that, despite the additional computational cost, CML exhibits superior training efficiency compared to MAML, converging faster and requiring less time to achieve the same accuracy goal.
>
> **[Q3]**
> Injecting noise into gradients has previously been proposed as a regularization method. [3] and [4] experimentally show that they have well-generalized parameters by injecting gradients generated from sub-networks or large networks into the original network. In the context of meta-learning frameworks that emphasize adaptation to a variety of tasks, it is important to have a well-generalized meta-initialization.
>
> Motivated by this, we propose a new meta-learning framework that combines multi-head frameworks to inject meaningful gradients. Our experimental results in Section 4.4 show that the co-learner's noise injections lead to more dynamic representation changes for new tasks, and CML$\dagger$ achieve high performance without adaptation. This suggests that the co-learner has the ability to inject a gradient that is sufficiently meaningful to the original gradient. As a result, the co-learner leads to convergence to well-generalized parameters that are adaptable to a variety of tasks.
>
> (Unfinished)

---

### Official Review · Reviewer_vYCb · 2024-03-22

**Q2-1 Originality-Novelty:** 2
**Q2-2 Correctness-Technical Quality:** 3
**Q2-5 Clarity Of Writing:** 1

**Q1 Summary And Contributions:**

The paper proposes CML, which extends model-agnostic meta-learning (MAML) for few-shot learning. Concretely, CML introduces a co-learner, which injects learnable noise into the meta-gradients and is updated in the outer-loop. During meta-testing, the CML is removed when fine-tuning to new tasks. The authors demonstrate the effectiveness of CML on regression, image classification, and node classification tasks. The authors also provide some insights by comparing with random gradient noise injection, visualizing the gradient similarity between the meta-learner and co-learner, etc.

**Q2-3 Extent To Which Claims Are Supported By Evidence:**

2: Fair: the main claims are somewhat supported by evidence (but the experimental evaluation may be weak, or does not match entirely with the claims, important baselines may be missing, proofs contain important ideas but lack rigor, algorithmic details are only discussed superficially, references are imprecise, assumptions are not sufficiently motivated or explicated, etc.).

**Q2-4 Reproducibility:**

3: Good: key resources (e.g. proofs, code, data) are available and key details (e.g. proofs, experimental setup) are sufficiently well-described for competent researchers to confidently reproduce the main results.

**Q3 Main Strengths:**

1. The paper introduces a novel approach to added a co-learner, which removes task-specific knowledge from the MAML system. It enables the MAML system to better learn shared meta-knowledge.
2. The authors design diverse experiments  to demonstrate its effectiveness.
3. The empirical provides some insights such as CKA and gradient similarity analysis.

**Q4 Main Weakness:**

2. It is not clear to me why CML can inject learnable noise into the gradient of the model. Based on my understanding, CML contains parameters that are fast updated during the inner loop and parameters that are slowly updated only in the outer loop. Are there any empirical or theoretical evidence that co-learner is injecting noise to the gradient? Please clarify.
3. The whole method section only mentions `noise’ once. If the co-learner injects noise to the MAML system. Please explain more how and why CML design can achieve the goal.
4. How are random noise added in Sec4.4? Are they added to a specific layer or all learnable parameters?
5. The sentence `Next, we demonstrate the convergence of the meta-gradient as a result of the co-learner’s gradient.’ summarizes the goal of the theorem. However, it is not a complete sentence and is hard to read.

**Q5 Detailed Comments To The Authors:**

See above sections for the strengths and weakness. Besides, the paper contains many grammar issues and is difficult to read. Please conduct proofreading for a few rounds.

**Q9 Complying With Reviewing Instructions:**

Yes

---

> ### Author Rebuttal · Authors · 2024-04-07
>
> We would like to thank the Reviewer vYCb for the detailed review of our work. We appreciate your pointing out not only the novelty of our proposed co-learner, but also the core of our paper, which is noise injection.
>
> **[Q1/Q2]**
> CML performs task-adaptation within the inner loop, similar to standard MAML, through the feature extractor and meta-learner while the co-learner is fixed. The co-learner intervenes in the outer loop, collaborating with the meta-learner to perform inference and encouraging the feature extractor to learn more diverse representations. Our co-learner is updated solely in the outer loop, learning general classification knowledge rather than task-specific knowledge. We believe a well-trained co-learner generates meaningful gradients during the update process, acting as beneficial noise to the original gradients.
>
> During backpropagation in CML, gradients from the last layer of the feature extractor are aggregated with the gradients from the meta-learner and co-learner. Let's denote the gradients from the meta-learner and co-learner as $G(\psi^{\prime}, \theta^{\prime})$ and $\bar{G}(\psi^{\prime}, \phi)$ respectively, as detailed in Theorem 1. Therefore, the gradient $G(\psi^{\prime}, \theta^{\prime}, \phi)$ of the feature extractor is calculated as follows:
> $G(\psi^{\prime}, \theta^{\prime}, \phi)$ =
> $G(\psi^{\prime}, \theta^{\prime})$ + $\bar{G}(\psi^{\prime}, \phi)$ =
> $(g{^{\mathcal\psi^\prime}_1}+\bar{g}{^{\mathcal\psi^\prime}_1}) + \cdots + (g{^{\mathcal\psi^\prime}_N}+\bar{g}{^{\mathcal\psi^\prime}_N}) + g^{\theta^{\prime}}$
>
> where $G(\psi^{\prime}, \theta^{\prime})$ is the same as the standard MAML framework. However, in the CML framework, a new noise gradient $\bar{G}(\psi^{\prime}, \phi)$ from the co-learner augments the original gradient to lead to a new gradient flow.
>
> we represent $\bar{G}(\psi^{\prime}, \phi)$ as the meaningful or learnable noise, similar to [1], because $\bar{G}(\psi^{\prime}, \phi)$ is calculated by learned co-learner at every outer loop.
>
> In the paper, Figure 3-(a) shows that the co-learner's gradient acts meaningfully, almost like a learnable gradient, promoting faster convergence during training. We also compared our $\bar{G}(\psi^{\prime}, \phi)$ with a random noise that is calculated by an untrained co-learner in Section 4.4.
>
> **[Q3]**
> We introduced a co-learner that does not perform updates (i.e., inner and outer loops) in the MAML framework to generate random noise. By doing so, the meta-gradient of MAML will be added as a randomized gradient to the original gradient flow by a non-updating co-learner. So, this random noise is automatically calculated and added to the original gradient during model updates.
>
> **[Q4]**
> Thank you for your feedback on the paper's readability. Our theory aims to converge theoretically when the gradients calculated from the co-learner are combined into a meta-gradient. Consequently, we have updated the paper to clarify this point.
>
> Thank you.
>
> Authors
>
> ---
>
> [1] GradAug: A New Regularization Method for Deep Neural Networks. NeurIPS 2018.

---

### Official Review · Reviewer_GHPV · 2024-03-23

**Q2-1 Originality-Novelty:** 2
**Q2-2 Correctness-Technical Quality:** 3
**Q2-5 Clarity Of Writing:** 3

**Q1 Summary And Contributions:**

This paper proposes a cooperative meta-learning (CML) framework for finding the better meta-initialization parameters with a plug-and-play module co-learner in meta-optimization phase. Experiments on few-shot regression, image classification and graph node classification show the superiority of the proposed method.

**Q2-3 Extent To Which Claims Are Supported By Evidence:**

3: Good: the main claims are supported by convincing evidence (in the form of adequate experimental evaluation, proofs, (pseudo-)code, references, assumptions).

**Q2-4 Reproducibility:**

3: Good: key resources (e.g. proofs, code, data) are available and key details (e.g. proofs, experimental setup) are sufficiently well-described for competent researchers to confidently reproduce the main results.

**Q3 Main Strengths:**

S1. Extending the traditional meta-learning framework MAML with a co-learner in meta-optimization is reasonable and can be easily incorporated with other MAML based methods.

S2. Theoretical proofs of the convergence of the co-learner’s gradient is provided.

S3. Extensive experiments on different few-shot learning tasks have been conducted and the results verify the effectiveness and efficiency of the proposed method.

**Q4 Main Weakness:**

W1. Although three few-shot learning tasks are tested in experiments, the datasets for each task are not enough, e.g., only the results of image classification on MiniImagenet are reported in Table 1, and only two datasets CoraFull and CiteSeer are used for graph node classification. More datasets should be adopted in the tests.

**Q5 Detailed Comments To The Authors:**

Q1. The presentation should be improved, e.g., the text in Fig. 2-3 and Table 2-3 is too small.

**Q9 Complying With Reviewing Instructions:**

Yes

---

> ### Author Rebuttal · Authors · 2024-04-07
>
> Thank Reviewer GHPV for providing detailed and helpful feedback and pointing out the innovativeness and contribution of our work. We appreciate all your comments, and we will reflect them in the final version.
>
> **[Q1]**
> For graph node classification, we conducted experiments by adding an Amazon computer datasets [1], and for image classification, we additionally trained on Tiered ImageNet [2]. Moreover, we further introduced a heavier backbone to demonstrate the superiority of CML. We report the results of our experiments in the **Global comment**.
>
> **[Q2]**
> We will increase the size of the text in Figure 2-(3) and Table 2-(3) as you requested. Additionally, in the final version, we will increase the size of the smaller text to enhance readability.
>
> Thank you.
>
> Authors
>
> ---
>
> [1] Pitfalls of Graph Neural Network Evaluation. NeurIPS 2018.
>
> [2] Meta-Learning for Semi-Supervised Few-Shot Classification. ICLR 2018.

---

### Official Review · Reviewer_ZVh4 · 2024-03-23

**Q2-1 Originality-Novelty:** 3
**Q2-2 Correctness-Technical Quality:** 4
**Q2-5 Clarity Of Writing:** 3

**Q1 Summary And Contributions:**

This paper proposes Cooperative Meta-Learning, a novel framework that introduces a co-learner module to induce gradient diversity and augment the meta-gradient in gradient-based meta-learning methods like MAML. The key idea is that the co-learner is only updated in the outer loop of meta-training and generates gradients from a different perspective than the main meta-learner. This gradient augmentation improves generalization. The co-learner can be removed at test time without impacting performance (compared to ANIL and BOIL).

**Q2-3 Extent To Which Claims Are Supported By Evidence:**

4: Excellent: all claims are supported by very convincing evidence (in the form of comprehensive experimental evaluation, rigorous mathematical proofs, detailed (pseudo-)code, precise references, well-motivated and realistic assumptions) and the authors deliver what they promise.

**Q2-4 Reproducibility:**

3: Good: key resources (e.g. proofs, code, data) are available and key details (e.g. proofs, experimental setup) are sufficiently well-described for competent researchers to confidently reproduce the main results.

**Q3 Main Strengths:**

1. In-depth experimental evaluation against SOTA meta learners.
2. Reduction of parameters can allow to train/utilize more complex models within a limited computational budget.
3. Detailed ablation studies and analysis of the method.

**Q4 Main Weakness:**

1. Evaluation on more challenging datasets can strengthen claims further
2. Algorithm 1 could be split up into 2 separate blocks to improve readability

**Q5 Detailed Comments To The Authors:**

None.

**Q9 Complying With Reviewing Instructions:**

Yes

---

> ### Author Rebuttal · Authors · 2024-04-07
>
> First of all, we sincerely appreciate the time and effort Reviewer ZVh4 has dedicated to reviewing our paper. We will endeavor to incorporate the comments you mentioned as much as possible into the final version of our paper.
>
> **[Q1]**
> We conducted additional experiments on graph data [1] and few-shot classification [2], and also included validation experiments on the heavier backbone ResNet12 [3]. We report the results of our experiments in our **Global comment**. We have made the code about the main results publicly available in supplementaries, and we will also upload the code for the additional experiments.
>
> **[Q2]**
> To enhance readability, we will divide Algorithm 1 into two parts, including meta training and meta testing.
>
> Thank  you.
>
> Authors
>
> ---
>
> [1] Pitfalls of Graph Neural Network Evaluation. NeurIPS 2018.
>
> [2] Meta-Learning for Semi-Supervised Few-Shot Classification. ICLR 2018
>
> [3] TADAM: Task dependent adaptive metric for improved few-shot learning. NeurIPS 2018.

---

### Meta-Review · Area_Chair_stLo · 2024-04-15

**Summary:** The paper introduces a regularization method for MAML where learnable noise is injected into the outer loop of MAML to get initializations for the inner loop that are more robust and generalize better. Since the cooperative noise injection can be removed after meta-training, the method incurs no additional cost at test time. Good empirical results support the usefulness and efficacy of the method.

**Recommendation:** After the rebuttal, all reviewers are in favor of accepting the paper. Unfortunately some reviewers were not responsive, but I personally think that some of their issues raised was addressed by the authors, so the actual score of the paper might be even higher. The paper generally scores high (3) in terms of clarity, reproducibility, evidence, etc., but most reviewers rate the novelty as a 2 only. Reviewers raised some important issues which were, in my opinion, well addressed by the authors. The only reviewer that was negative about the paper pre-rebuttal changed their score to a 6 after rebuttal. Taking all of this information together, I recommend accepting the paper as a poster, leaning mildly towards Spotlight.